# Single-cell radioluminescence microscopy with two-fold higher sensitivity using dual scintillator configuration

**Tae Jin Kim**[1]*, Qian Wang[2], Mark Shelor[3], Guillem Pratx[1]

1 Department of Radiation Oncology, Stanford School of Medicine, Stanford, California, United States of America, 2 Department of Bioengineering, University of California, Davis, California, United States of America, 3 Department of Biomedical Engineering, University of California, Merced, California, United States of America

* tj_kim@utexas.edu

**Data Availability Statement:** All relevant data are within the paper.

## Abstract

Radioluminescence microscopy (RLM) is an imaging technique that allows quantitative analysis of clinical radiolabeled drugs and probes in single cells. However, the modality suffers from slow data acquisition (15–30 minutes), thus critically affecting experiments with short-lived radioactive drugs. To overcome this issue, we suggest an approach that significantly accelerates data collection. Instead of using a single scintillator to image the decay of radioactive molecules, we sandwiched the radiolabeled cells between two scintillators. As proof of concept, we imaged cells labeled with [18F]FDG, a radioactive glucose popularly used in oncology to image tumors. Results show that the double scintillator configuration increases the microscope sensitivity by two-fold, thus reducing the image acquisition time by half to achieve the same result as the single scintillator approach. The experimental results were also compared with Geant4 Monte Carlo simulation to confirm the two-fold increase in sensitivity with only minor degradation in spatial resolution. Overall, these findings suggest that the double scintillator configuration can be used to perform time-sensitive studies such as cell pharmacokinetics or cell uptake of short-lived radiotracers.

## Introduction

Positron emission tomography (PET) is a molecular imaging technique that enables interrogation of biophysical processes in living subjects in a non-invasive manner. It is popularly used in the clinic to diagnose and characterize various diseases such as cancer, cardiovascular disorders, and neurological disorders using a wide range of radiotracers [1–4], including radioactive glucose 2-deoxy-2-[18F]fluorodeoxyglucose or [18F]FDG. [18F]FDG is widely used in the clinic for detecting and staging cancer [5–8].

Due to their clinical significance, novel radiopharmaceuticals are actively investigated for diagnostic and therapeutic purposes. However, the biological activity of radiopharmaceuticals is difficult to confirm—the spatial resolution of current clinical and pre-clinical imaging systems is limited to the tissue level. This makes it difficult to acquire detailed information of how

**Funding:** G.P. was supported by funding from the National Institutes of Health (NIH) under grant 5R01CA186275. T.J.K. was supported in part by NCI training grant T32CA118681. The funders had no role in study design, data collection and analysis, decision to publish, or preparation of the manuscript.

**Competing interests:** The authors have declared that no competing interests exist.

these radioactive molecules interact with target cells that present heterogeneous molecular characteristics. Techniques that enable single-cell radionuclide detection (e.g., micro-autoradiography [9,10]) are challenging to implement and not applicable to live cells, and are thus rarely used.

Radioluminescence microscopy (RLM) was introduced to enable *in vitro* radionuclide imaging of live single cells. In RLM, a scintillator crystal is placed directly under or above the radiolabeled cells to observe optical flashes resulting from the decay of single radioactive molecules. By capturing a series of image frames and individually counting the scintillation flashes within each frame, RLM can quantify how many radioactive molecules are present within individual cell with high sensitivity–down to fewer than 1,000 molecules per cell. The unique capabilities of RLM have revealed previously unattainable information on single-cell response to clinical radiotracers [11–13]. However, one limitation of this modality is the relatively long image acquisition time, which ranges from 15 to 30 minutes per sample.

In this study, we introduce a simple method to significantly reduce image acquisition time. Compared to the original approach, which uses a single scintillator to capture radioluminescence signals, we sandwich the radiolabeled cells between two scintillators. This allows complete geometric coverage for increased detection efficiency. As proof-of-concept, human breast cancer cells (MDA-MB-231) grown on a cadmium tungstate ($CdWO_4$) scintillator are incubated with [18F]FDG and imaged with both single and dual scintillator configurations. The experimental results are further compared with Monte Carlo simulations to demonstrate that the detection sensitivity of RLM doubles when using two scintillators.

Since the time required to acquire radioluminescence images can be considerably reduced, potential applications of the technique include identifying radiolabeled histological tumor samples from patients and screening the efficacy of novel radiopharmaceuticals in biological specimens.

## Materials and methods

### Sample preparation

**Cell culture.** A cadmium tungstate scintillator ($CdWO_4$, MTI Co.) with a dimension of $10 \times 10 \times 0.5$ mm (width × length × height) was coated with 100 μl of fibronectin (5 μg/ml) to promote cell adhesion to the crystal surface. The scintillators were incubated for ~2 hours and washed three times with sterile distilled water. The treated scintillator was placed on a 35 mm diameter cover-slip bottom dish (μ-Dish, ibidi GmbH), and MDA-MB-231 human breast cancer cells suspended in DMEM were dispensed in the dish with cell density of $10^4$ cells/ml. The cells were then incubated in a $CO_2$ incubator for 24 h prior to the experiment to allow them to attach to the scintillator surface.

**Radiolabeling cells.** The MDA-MB-231 cells were labeled with radioactive glucose analogue [18F]FDG, a radiotracer that is commonly used in the clinic to detect and stage cancer in patients. The cells were first incubated in glucose-free DMEM for 30–40 minutes. The culture media was then replaced with a solution of glucose-free DMEM containing 22–24 KBq/ml of [18F]FDG, and the cells were incubated for an additional 45 minutes. After the radiolabeling process, the cells were washed with clear glucose-free DMEM.

### Radionuclide imaging

Radioluminescence imaging was performed with a low-light microscope developed in-house [14]. The microscope consists of a 20× / 0.75 NA microscope objective lens (Nikon, CFI Plan Apo Lambda) coupled to a 36 mm tube lens. This yields an effective magnification of 3.6× while maintaining the native numerical aperture of the objective lens [15]. The microscope is

mounted on top of a highly sensitive EMCCD camera (C9100-13, Hamamatsu Co.). To protect the EMCCD camera from stray light and to minimize the background signal, the entire microscope was enclosed in a customized light-tight box.

The culture dish containing the scintillator with MDA-MB-231 cells was then placed on the microscope stage. A brightfield image was first captured with the camera settings set to standard mode (non-EMCCD mode). The first RLM image was then acquired (Fig 1A), with the camera parameters set to 1,060 EM gain, 4 × 4 binning, 30 ms exposure time, and 10,000 acquisition frames. After the first set of images, another scintillator was gently placed on top of the cells (Fig 1B). A second set of RLM images was captured with identical camera parameters. After both images were acquired, the culture dish was removed from the microscope stage and a separate dark reference sequence was captured (1,000 frames) for background subtraction.

The radioactive decay signals were analyzed using the ORBIT toolbox (optical reconstruction of the beta-ionization track) [16]. In ORBIT, the reconstruction process starts by subtracting the background noise from each RLM image. Individual scintillation flashes are then isolated within each image frame and converted into $(x, y)$ event coordinates. Once this process is repeated for the entire set of 10,000 images, the positions of all detected events are aggregated into a single image, where each pixel represents the number of radioactive decay events detected at that location. Detailed reconstruction procedures and radioluminescence imaging can also be found from our previous papers [14,16,17].

## Monte Carlo simulation

Experimental results were also compared by simulating the radioluminescence process using a Monte Carlo software package (Geant4). Similar to our previous work [18,19], a single $^{18}$F point source with 1 Bq of radioactivity was generated in virtual space to represent a single radiolabeled cell. An optical model was implemented by convoluting the simulated scintillation tracks with the 3D point-spread function of the microscope's objective, then applying pixel noise characteristic of the EMCCD camera. Radioluminescence images were then reconstructed by selecting and counting the scintillation events occurring from the beta particles.

For the single scintillator experiment, the source was positioned 5 μm above a 100 μm-thick $CdWO_4$ slab. For the double scintillator configuration, a second $CdWO_4$ scintillator was placed 5 μm above the source, i.e. with the $^{18}$F point source equidistant from the two scintillators. The 5 μm offset corresponds to the height of the MDA-MB-231 cells assuming an average height of 10 μm, as previously observed [19]. A total of 20,000 radioactive decay events were simulated, and the corresponding radioluminescence images were generated. The raw images were then reconstructed using the same methods from the previous section with the ORBIT toolbox.

It should be noted that the thickness of the scintillator in the model was set to 100 μm for computational efficiency. Since the optical depth of field is ~24.5 μm, this thickness is sufficient for simulating the scintillation signals captured with the low-light microscope.

## Results

### Radioluminescence images of single vs. double scintillator

As expected, experimental results demonstrate that the double scintillator configuration yields significantly higher radioactive decay count. RLM images acquired with the two configurations are shown in Fig 2A and 2B, respectively. The figures are shown in color using the same intensity scale, where red represents higher radioactive decay counts measured at each pixel. Also, the double scintillator configuration can detect radioactivity from cells that are otherwise undetectable using only one scintillator (highlighted circles in Fig 2C and 2D). It should be

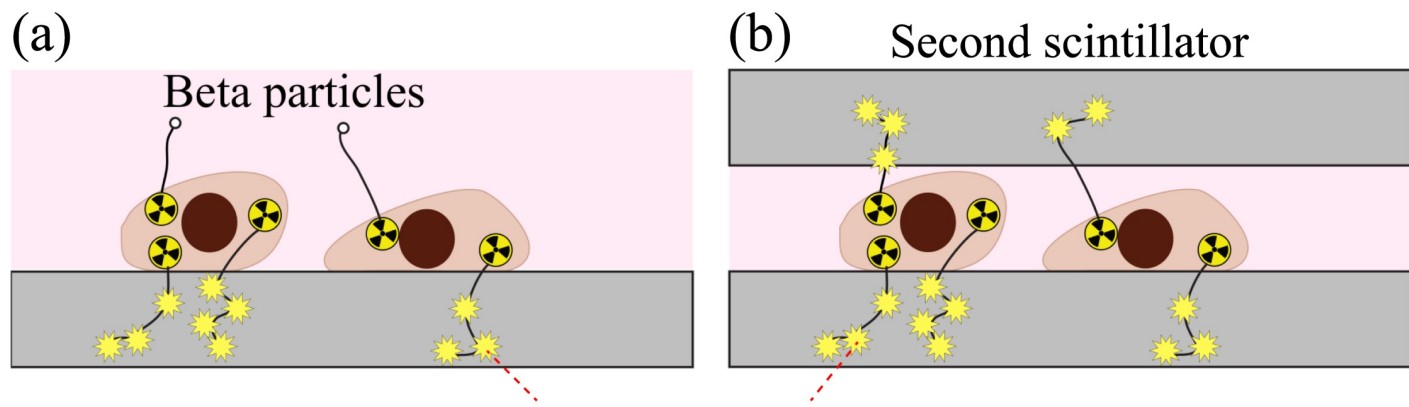

**Fig 1. Schematic diagrams.** Radioluminescence microscopy with (a) single scintillator and (b) double scintillator configurations.

noted that the radioactive decay counts of the double scintillator case were adjusted to correct for the half-life of $^{18}$F ($\tau_{1/2}$ ~110 minutes). Since the double scintillator image was acquired 21 minutes after the single scintillator image, the displayed count values were increased by 14%.

## Sensitivity comparison

The number of radioactive decay counts detected for each cell was estimated by drawing a circular region of interest (ROI; diameter, 90 μm) around individual cells ($N = 66$) and summing the number of decay counts within each ROI. Each ROI measurement was then corrected by subtracting a background signal, which was obtained by averaging the radioactive decay counts detected in empty areas devoid of cells ($N = 66$ ROIs).

The number of decay events measured in a single cell, defined as $D$, is related to the number of [$^{18}$F]FDG molecules $N_0$ within that cell at a reference time point. The relation is expressed as,

$$D = SYN_0 \left( 1 - \exp\left( -\frac{\ln 2}{\tau_{1/2}} t \right) \right) \tag{1}$$

where $S$ is the detection sensitivity of the RLM system, $Y$ is the radioactive yield for particulate radiation (0.97 for $^{18}$F), $\tau_{1/2}$ is the half-life (~110 minutes for [$^{18}$F]FDG), and $t$ is the elapsed time. The sensitivity increase can thus be expressed in terms of the ratio,

$$\frac{S_{\text{double}}}{S_{\text{single}}} = \frac{D_{\text{double}} \cdot \exp\left( -\frac{\ln 2}{\tau_{1/2}} \Delta t \right)}{D_{\text{single}}} \tag{2}$$

where $\Delta t = t_{\text{double}} - t_{\text{single}}$ is the time delay between the two measurements. The exponent term represents the decay correction due to time delay between the two experiments (and is equal to 1.14 for $\Delta t = 21$ min). This allows us to assume that the two measurements are simultaneously performed on the same ROI. The increase in sensitivity was computed both for background ROIs and ROIs containing single cells. For background signals (ROIs devoid of cells), we individually quantified the radioluminescence signals (Fig 3A). While the data scatter is relatively large ($r^2 > 0.69$), results demonstrate that the double scintillator configuration is indeed more sensitive than the single scintillator by a factor of 2. The average uptake values also show

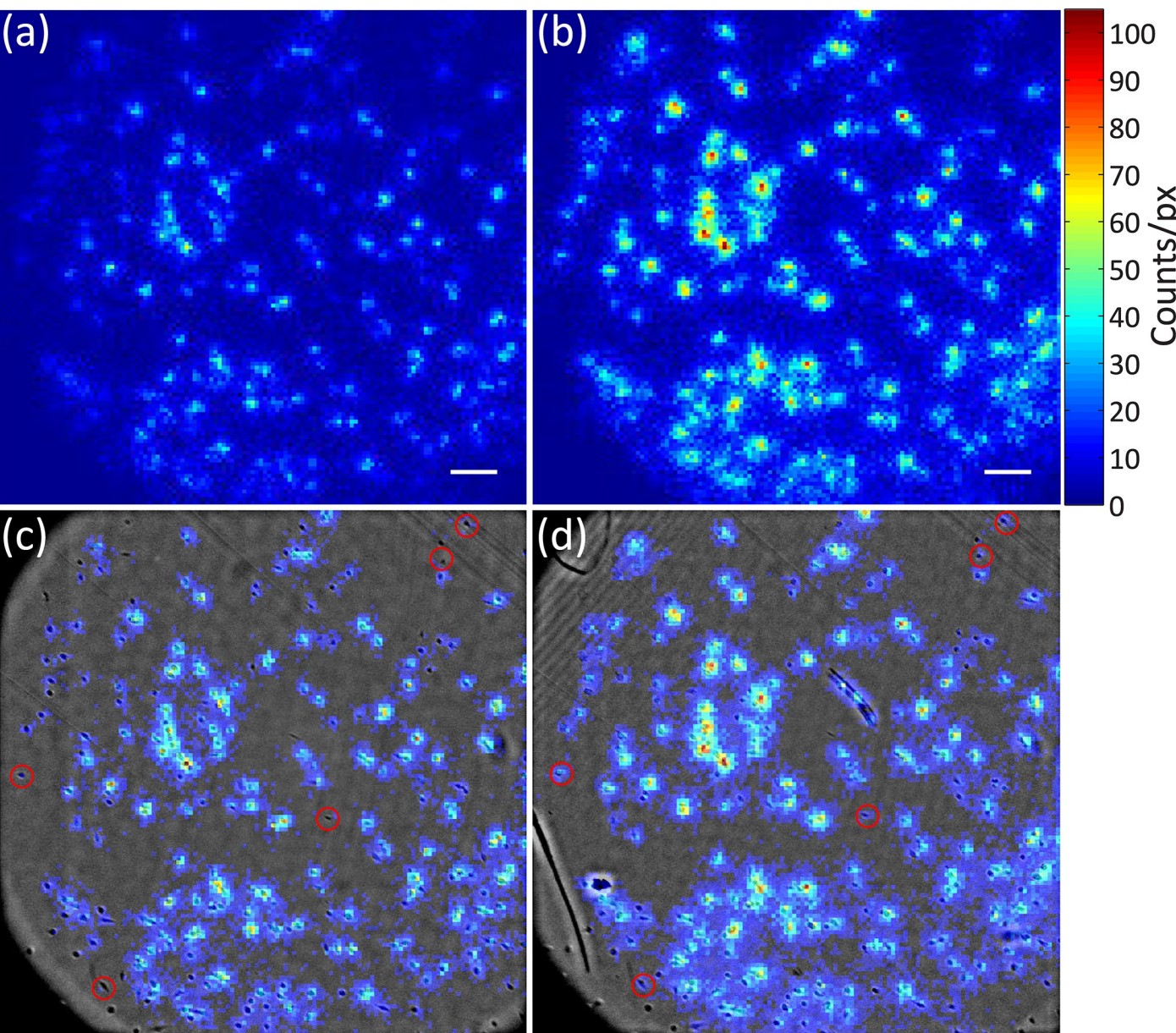

**Fig 2. RLM images of [¹⁸F]FDG uptake by MDA-MB-231 cells.** Raw RLM data acquired with (a) single scintillator and (b) double scintillator configuration. Composite image of radioluminescence and brightfield image of (c) single scintillator and (d) double scintillator. Red circles represent cells that are detectable with double scintillator only. Scale bar, 200 μm.

a two-fold increase in sensitivity, $73 \pm 34$ counts/ROI (mean ± one standard deviation) and $37 \pm 14$ counts/ROI for double and single scintillator, respectively.

   A comparison of radioactive decay counts detected per cell between single and double scintillator configurations are shown in Fig 3B. As expected, [¹⁸F]FDG uptake demonstrated a heterogeneous behavior, with some cells taking up the radiotracer 3–4 times more than other cells. Furthermore, the measured average cell uptake was increased by two-fold when the second scintillator was added. The average uptake values were $407 \pm 156$ counts/cell for the double scintillator configuration and $200 \pm 90$ counts/cell for the single scintillator case. Linear

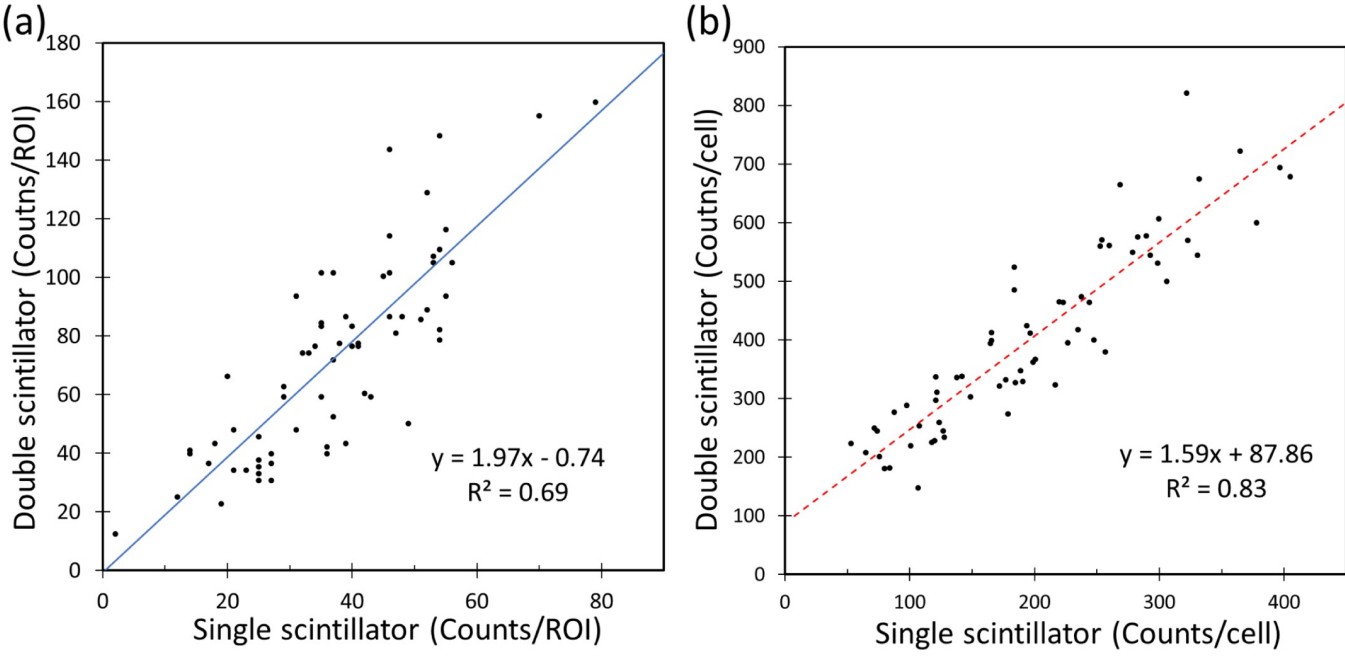

**Fig 3. Graphs representing single vs. double scintillator results.** Scatter plot of decay counts detected in 90 μm ROI ($N = 66$) for (a) background devoid of cells and (b) areas containing single cells. The solid blue line and dotted red lines represent the linear regression fit.

regression (dotted red line) of the two datasets found a linear relationship between the two datasets, $y = 1.59\,x + 87.86$ ($r^2 > 0.83$).

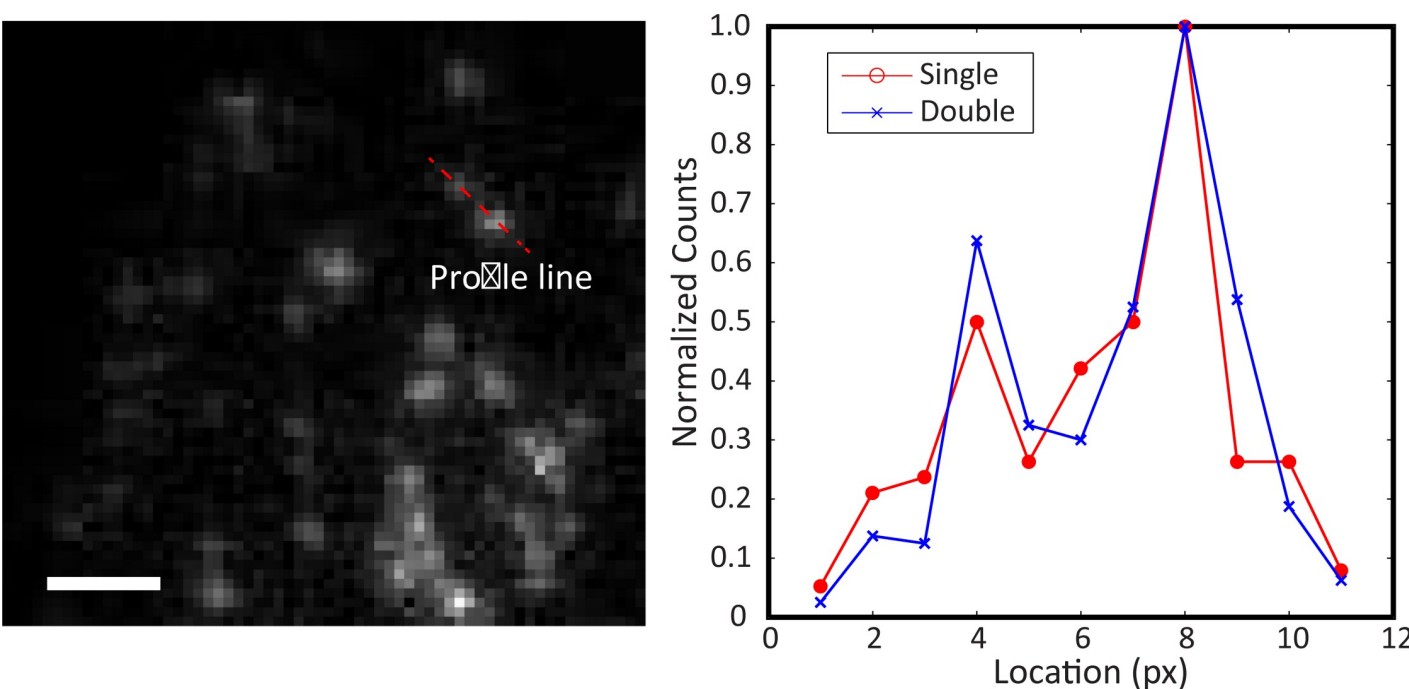

**Fig 4. Radioactive decay profile of two closely positioned cells.** (a) A profile is drawn between two radioactive cells in the radioluminescence image. The image is represented in grayscale for presentation. Scale bar 200 μm. (b) Graph of decay count profiles comparing radioluminescence image captured with single scintillator and double scintillator. Each plot is normalized by their respective maximum count values.

## Spatial resolution

The spatial resolution of the single and dual scintillator cases was quantitatively assessed by drawing a line profile across two radioactive cells (Fig 4A). The profile was normalized according to the maximum count values for both configurations (Fig 4B). Results clearly show two peaks separated by a valley, which correspond to two cells with [18F]FDG uptake along the line profile. Furthermore, no qualitative discrepancies in peak, valley, and background values can be observed between the two configurations. This simple comparison suggests that the sensitivity of RLM can be increased without significant degradation in spatial resolution.

To further assess the effects on spatial resolution, a Monte Carlo simulation of the system was implemented using the GEANT4 package. A single 18F point source (1 Bq) was simulated for both configurations. The point-spread function measured by using the single and double scintillator configuration is shown in Fig 5A and 5B, respectively. The simulation results agree well with the experimental results–the double scintillator is more sensitive than the single scintillator system by a factor of two. It can be seen from the simulation, however, that the single scintillator system provides slightly higher spatial resolution than the double scintillator system. Using a circularly symmetric 2-D Gaussian curve fit, we estimate the FWHM of both systems and find that the single-scintillator system achieves a spatial resolution of 44 μm, compared with 53 μm for the double-scintillator system.

## Discussion

The main finding of this study is that the double scintillator configuration increases the detection sensitivity by a factor of two compared to the single scintillator case. While there are clear advantages of using the double scintillator configuration over a single scintillator, a few considerations must be made. The first is the depth of field of the microscope, which must be large enough to accommodate both scintillators. The depth of field $d_{LLM}$ of a microscope is,

$$d_{LLM} = \frac{n \cdot \lambda}{NA^2} + \frac{n \cdot b \cdot e}{M \cdot NA}, \tag{3}$$

where $\lambda$ = 475 nm is the emission peak of the scintillation light, $n$ is the refractive index of the medium, $NA$ is the numerical aperture of the objective lens, $e$ is the pixel size, $b$ is the binning number, and $M$ is the effective magnification. As explained in a previous publication [15], the low-light microscope equipped with a 20× Nikon CFI Plan Apochromat λ has the following characteristics: $M_{effective}$ = 3.6, $n$ = 1, and $NA$ = 0.75. Considering the parameters of the EMCCD camera (pixel size of 16 μm × 16 μm and binning of 4 × 4), $d_{LLM}$ is estimated to be ~24.5 μm. Assuming that the distance between the bottom and top scintillator is ~ 10 μm [19], the depth of focus is sufficiently large to simultaneously focus on top and bottom edges of the scintillators. Since most of the scintillation signals is emitted near the surface of the scintillator, with an average penetration depth of the beta particles estimated as 25 μm, this depth of field is sufficient to capture a meaningful number of events from both scintillators. However, this is not true for all objective magnifications. For instance, a 40×/1.3 NA oil lens may not be compatible with the double-scintillator configuration. While $M_{effective}$ will increase to 7.2, $d_{LLM}$ will be reduced by half or ~12.3 μm. This depth of field may not be enough to simultaneously capture the radioluminescence signals from both scintillators.

It should be pointed out that there is [18F]FDG efflux from the live cells, a well-known phenomenon of the glucose analog radiotracer [20,21]. Since the double scintillator image was captured 21 minutes after the single scintillator configuration, [18F]FDG efflux can be visualized by directly comparing the two RLM images. The pixel values of the double scintillator image were divided by two to match the single scintillator sensitivity, and the image was then

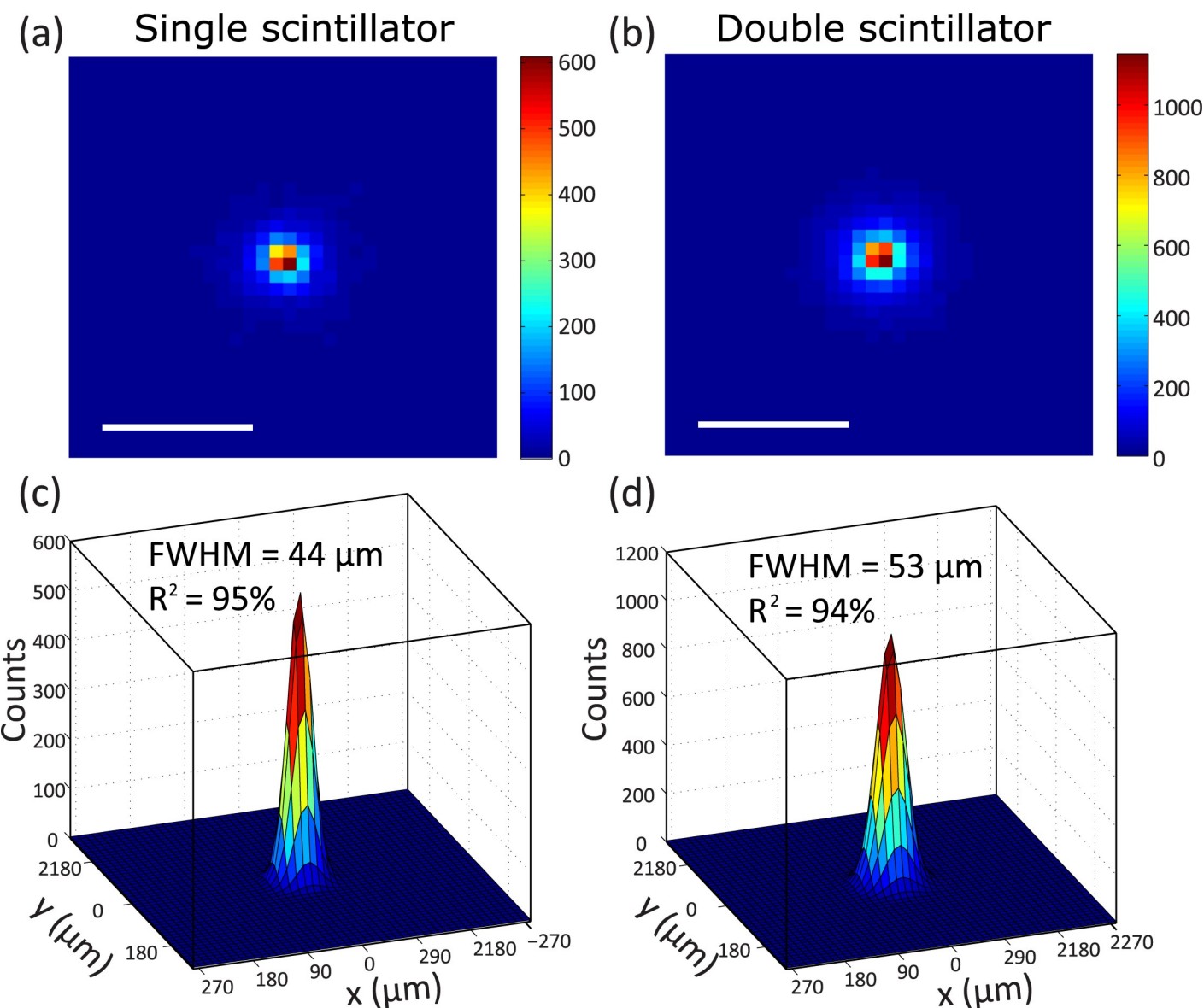

**Fig 5. Monte Carlo simulation results of a single radioactive point source.** Simulated RLM image of point source and corresponding 2D Gaussian fits with the (a, c) single and (b, d) double scintillator, respectively. The scintillation from 511 keV photons, which are photons produced from electron-positron annihilation events, were also included in the simulation but were negligible since they have a smaller probability than positrons to deposit energy within the ~25 μm-thick region of the scintillator that is in focus in the microscope. Scale bar, 200 μm.

subtracted by the single scintillator image. The resulting difference image is shown as Fig 6. In the figure, a significant number of radiolabeled cells with decreased activity are observed (shown in blue). On the other hand, there is a relative increase in radioactivity at the immediate vicinity of the cells. This is a clear indication that biological [18F]FDG efflux occurred between the two measurements, which also explains why the linear regression coefficient was less than ×2 in Fig 3B ($y = 1.59\,x + 87.86$).

Another confounding factor is that the double scintillator configuration may have increased signal contamination due to stray gamma rays, which may cause scintillation far away from the cell of origin. This may degrade the quality of radioluminescence images, particularly in

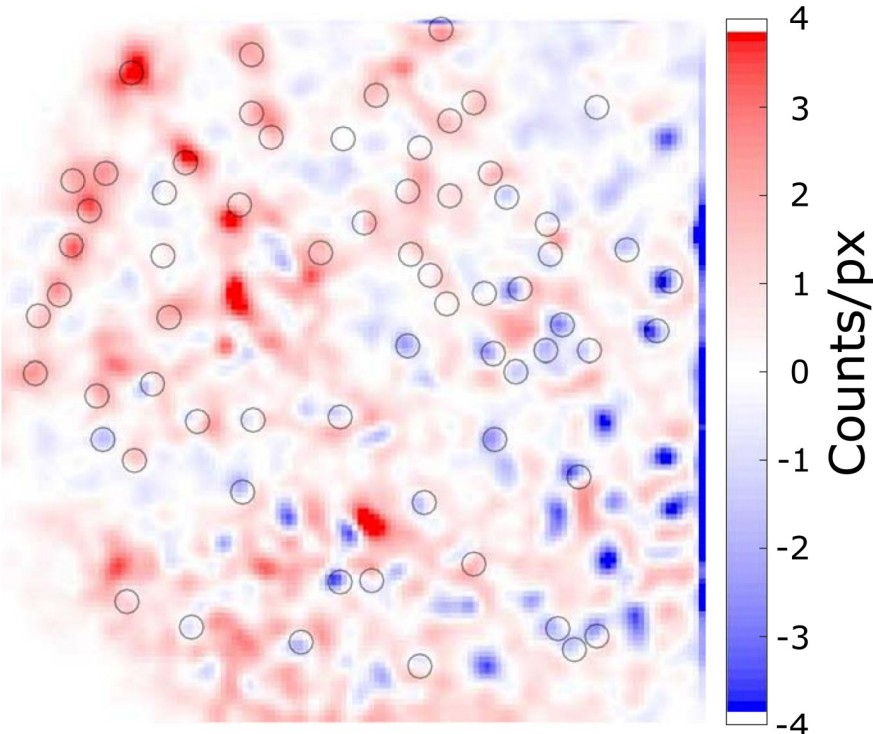

**Fig 6. Differential RLM image comparing double and single scintillator configurations.** The decay-corrected double scintillator image was divided in half, followed by subtracting the single scintillator result. The black circles are ROIs around single cells.

areas of high cell density. Further studies regarding the effect of radiation contamination from closely positioned cells is required.

## Conclusion

We demonstrated a method to double the sensitivity of radioluminescence microscopy or RLM. While RLM provides quantitative information on the biological response of tissues and cells to radiopharmaceuticals, the technique is limited by slow data acquisition time. To resolve this issue, we sandwiched the radiolabeled cells between two parallel scintillators. Results show that the sensitivity increased by a factor of two compared to the original single scintillator method, thus effectively reducing the acquisition time by half. While simulation results show that the double scintillator geometry has slightly lower spatial resolution, this effect was not noticeable during experimental cell imaging. Moreover, we were able to visualize radiotracer efflux around the cells, suggesting a new application that our technique can be used to simultaneously quantify radiotracer uptake and efflux. In conclusion, the significant reduction in image acquisition time will increase the experimental throughput and enabling pharmacokinetic analysis of short-lived clinical radiotracers. In addition, the dual scintillator configuration may be a suitable candidate for miniaturized radiobioassay devices, which is an emerging technology that enables evaluation of biological responses to novel radiotracers with minimal volume requirements [22].

## Author Contributions

**Conceptualization:** Tae Jin Kim, Guillem Pratx.

**Data curation:** Tae Jin Kim, Guillem Pratx.

**Formal analysis:** Tae Jin Kim, Qian Wang.

**Funding acquisition:** Tae Jin Kim, Guillem Pratx.

**Investigation:** Tae Jin Kim, Mark Shelor.

**Methodology:** Qian Wang, Mark Shelor.

**Software:** Tae Jin Kim, Qian Wang.

**Supervision:** Guillem Pratx.

**Visualization:** Tae Jin Kim, Guillem Pratx.

**Writing – original draft:** Tae Jin Kim, Guillem Pratx.

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
