## [Decision Letter · Decision Letter 0]

4 May 2020

PONE-D-19-21530

Single-cell radioluminescence microscopy with two-fold higher sensitivity using dual scintillator configuration

PLOS ONE

Dear Dr. Kim:

Thank you for submitting your manuscript to PLOS ONE. After careful consideration, we feel that it has merit but does not fully meet PLOS ONE’s publication criteria as it currently stands. Therefore, we invite you to submit a revised version of the manuscript that addresses the points raised during the review process.

Comments from the reviewers are listed below.

We would appreciate receiving your revised manuscript by Jun 15 2020 11:59PM. To enhance the reproducibility of your results, we recommend that if applicable you deposit your laboratory protocols in protocols.io, where a protocol can be assigned its own identifier (DOI) such that it can be cited independently in the future. For instructions see: http://journals.plos.org/plosone/s/submission-guidelines#loc-laboratory-protocols

We look forward to receiving your revised manuscript.

Kind regards,

Gayle E. Woloschak, PhD

Academic Editor

PLOS ONE

Additional Editor Comments:

Journal Requirements:

2. In your Methods section, please provide additional details regarding the cell lines used in your study and ensure you have described the source. For more information regarding PLOS' policy on materials sharing and reporting, see https://journals.plos.org/plosone/s/materials-and-software-sharing#loc-sharing-materials, and for more information on PLOS ONE's guidelines for research using cell lines, see https://journals.plos.org/plosone/s/submission-guidelines#loc-cell-lines.

One reviewer rejected the work, the other accepted it. Please address as many comments as possible from the reviewers in the revision.

Reviewers' comments:

Reviewer's Responses to Questions

**Comments to the Author**

1. Is the manuscript technically sound, and do the data support the conclusions?

Reviewer #1: Yes

Reviewer #2: Partly

2. Has the statistical analysis been performed appropriately and rigorously? 

Reviewer #1: Yes

Reviewer #2: No

3. Have the authors made all data underlying the findings in their manuscript fully available?

Reviewer #1: Yes

Reviewer #2: Yes

4. Is the manuscript presented in an intelligible fashion and written in standard English?

Reviewer #1: Yes

Reviewer #2: Yes

5. Review Comments to the Author

Reviewer #1: This is a clearly written paper describing a straightforward modification to the authors' previously developed radioluminescence microscopy setup. In this imaging modality, cells are grown on a scintillator and treated with a beta-emitting radionuclide. The beta emissions cause scintillations that can be detected with a custom optimized low-light microscope that allows for identification of individual cells and quantification of their affinity for the radionuclide. However, the system is very much quantum limited and imaging takes 20-30 minutes.

In this paper, the authors simply place a second scintillator above the cells in an effort to double the system sensitivity and halve the imaging time. They show, with experiment and simulation, that this is indeed achieved, with little downside, except some potential small loss in resolution.

The loss of resolution is due to the fact that the scintillator sandwich can exceed the depth of focus of the microscope, leading portions of the scintillators to be imaged at lower than optimal resolution.

The only two (related) points I think could be clarified are:

The experimental scintillator is 500 microns thick but the simulated one is 100 microns thick. Why are these not the same?

Both of these scintillator thicknesses will lead to a “sandwich" that is greatly in excess of the calculated 25 micron depth of field. Is it assumed that the betas all interact near the proximal surfaces of the two scintillators? What is the expected range of the betas in CdW4?

Reviewer #2: Comments to Authors: This study investigated an accelerated process to image human breast cancer cells MDA-MB-231 that was imaged with two scintillator slabs after incubated with [18F]FDG. Although the authors claimed to improve the sensitivity to 2-fold and cut the data acquisition time into half, I am not convinced that this signal increase is due to positron emission. This study used two CdWO4 scintillators that sandwiched these MDA-MB-231 cancer cells. As each scintillator is 0.5 mm in thickness, therefore together they have 1 mm thickness, that would have reduced intrinsic efficiency as low as 10% due to anhelation photons. Then how this is possible to have sensitivity increased by 2-fold specifically coming from positron emission. The authors did not address this issue how they account for these photons in the Monte Carlo simulation. Also, in the Monte Carlo Simulation source was positioned 5 µm above a 100 um thick CdWO4 slab. It is not clear from the why this space was added. It may be beneficial to clarify this aspect. the study is interesting, Also, the study is failed to show clinical pull for this imaging technique. It will be helpful to add a new paragraph in the Introduction section.

6. PLOS authors have the option to publish the peer review history of their article (what does this mean?). If published, this will include your full peer review and any attached files.

Reviewer #1: No

Reviewer #2: No

---

## [Author Response · Author response to Decision Letter 0]

16 Jun 2020

The authors uploaded the reviewer response in PDF form since there are figures to support our comments.

---

## [Decision Letter · Decision Letter 1]

22 Jun 2020

Single-cell radioluminescence microscopy with two-fold higher sensitivity using dual scintillator configuration

PONE-D-19-21530R1

Dear Dr. Kim:

We’re pleased to inform you that your manuscript has been judged scientifically suitable for publication and will be formally accepted for publication once it meets all outstanding technical requirements.

Kind regards,

Gayle E. Woloschak, PhD

Section Editor

PLOS ONE

Additional Editor Comments (optional):

Thank you for addressing the concerns of the reviewers.

Reviewers' comments:

Reviewer's Responses to Questions

**Comments to the Author**

1. If the authors have adequately addressed your comments raised in a previous round of review and you feel that this manuscript is now acceptable for publication, you may indicate that here to bypass the “Comments to the Author” section, enter your conflict of interest statement in the “Confidential to Editor” section, and submit your "Accept" recommendation.

Reviewer #1: All comments have been addressed

Reviewer #2: All comments have been addressed

2. Is the manuscript technically sound, and do the data support the conclusions?

Reviewer #1: Yes

Reviewer #2: Yes

3. Has the statistical analysis been performed appropriately and rigorously? 

Reviewer #1: Yes

Reviewer #2: Yes

4. Have the authors made all data underlying the findings in their manuscript fully available?

Reviewer #1: Yes

Reviewer #2: Yes

5. Is the manuscript presented in an intelligible fashion and written in standard English?

Reviewer #1: Yes

Reviewer #2: Yes

6. Review Comments to the Author

Reviewer #1: The responses to my questions are very clear and enlightening. The same is true for the responses to the other reviewer.

Reviewer #2: The authors of this manuscript fully addressed all the questions and comments from the previous review.

7. PLOS authors have the option to publish the peer review history of their article (what does this mean?). If published, this will include your full peer review and any attached files.

Reviewer #1: Yes: Patrick La Riviere

Reviewer #2: No

---

## [Editor Report · Acceptance letter]

26 Jun 2020

PONE-D-19-21530R1 

Single-cell radioluminescence microscopy with two-fold higher sensitivity using dual scintillator configuration 

Dear Dr. Kim:

I'm pleased to inform you that your manuscript has been deemed suitable for publication in PLOS ONE. Congratulations! Your manuscript is now with our production department. 

Kind regards, 

on behalf of

Dr. Gayle E. Woloschak 

Section Editor

PLOS ONE